# Systems Metabolic Engineering of *Saccharomyces cerevisiae* for the High-Level Production of (2*S*)-Eriodictyol

**DOI:** 10.3390/jof10020119

**Published:** 2024-01-31

**Authors:** Siqi Zhang, Juan Liu, Zhiqiang Xiao, Xinjia Tan, Yongtong Wang, Yifei Zhao, Ning Jiang, Yang Shan

**Affiliations:** 1Longping Branch, College of Biology, Hunan University, Changsha 410125, China; 18861822835@163.com (S.Z.); liujmax2019@163.com (J.L.); xiaozhiqiang@hnu.edu.cn (Z.X.); tanxinjia0209@163.com (X.T.); quarknorthstar@163.com (Y.W.); zhaoyifei@hnu.edu.cn (Y.Z.); jiangning0806@outlook.com (N.J.); 2Agriculture Product Processing Institute, Hunan Academy of Agricultural Sciences, Changsha 410125, China; 3Hunan Key Lab of Fruits & Vegetables Storage, Processing, Quality and Safety, Hunan Agricultural Products Processing Institute, Changsha 410125, China; 4Department of Life Sciences, Chalmers University of Technology, SE412 96 Gothenburg, Sweden

**Keywords:** flavonoid, (2*S*)-naringenin, (2*S*)-eriodictyol, metabolic engineering, metabolic balance

## Abstract

(2*S*)-eriodictyol (ERD) is a flavonoid widely found in citrus fruits, vegetables, and important medicinal plants with neuroprotective, cardioprotective, antidiabetic, and anti-obesity effects. However, the microbial synthesis of ERD is limited by complex metabolic pathways and often results in a low production performance. Here, we engineered *Saccharomyces cerevisiae* by fine-tuning the metabolism of the ERD synthesis pathway. The results showed that the ERD titer was effectively increased, and the intermediate metabolites levels were reduced. First, we successfully reconstructed the de novo synthesis pathway of *p*-coumaric acid in *S. cerevisiae* and fine-tuned the metabolic pathway using promoter engineering and terminator engineering for the high-level production of (2*S*)-naringenin. Subsequently, the synthesis of ERD was achieved by introducing the *ThF3*′*H* gene from *Tricyrtis hirta*. Finally, by multiplying the copy number of the *ThF3*′*H* gene, the production of ERD was further increased, reaching 132.08 mg L^−1^. Our work emphasizes the importance of regulating the metabolic balance to produce natural products in microbial cell factories.

## 1. Introduction

Flavonoids are a class of phenylpropanoid compounds with a basic structure consisting of a C6-C3-C6 carbon skeleton [1]. Most flavonoids have antiviral, antioxidant, and anti-inflammatory properties [2]. Equally, flavonoids have potential applications in the control of lung diseases, especially acute respiratory distress syndrome caused by COVID-19 [3]. (2*S*)-eriodictyol (ERD) is a flavanone which has highly analgesic [4], antioxidant [5] and anti-inflammatory [6], antipyretic [7] and antinociceptive actions [8], and antitumor activity [9], so ERD is a potent source of natural remedies for maintaining a high standard of health [10].

Over the past few decades, with rapid advances in metabolic engineering and synthetic biology, microbe-based bioproduction has increasingly become an alternative to traditional extraction from plants or chemical synthesis [11]. By re-engineering the cellular metabolism of fast-growing microorganisms (e.g., *Escherichia coli* and *Saccharomyces cerevisiae*), artificial cellular platforms have been successfully constructed to produce high levels of chemicals ranging from biofuels to proteins [12]. The production of heterologous proteins in microorganisms is limited by the characteristics of the enzymes and differences in the expression systems [13], leading to problems such as a low substrate conversion rate, the substantial accumulation of intermediates, and poor yields/titers of the final products. Nowadays, many strategies have been employed to address these problems at different levels and in different ways, such as gene transcription and translation, protein folding, and enzyme modification. At the gene expression level, to produce the target products efficiently, the balance of metabolic flux is usually regulated by establishing and screening promoters with different strengths [14]. For protein folding and enzyme modification, strategies such as the co-expression of molecular chaperones, truncation of the endoplasmic reticulum membrane-anchored peptide, and the addition of fusion tags are effective in promoting the proper protein folding, as well as the soluble expression and stability of enzymes [15,16,17]. Other strategies, such as increasing the copy number of genes encoding rate-limiting enzymes, weakening the competing pathways, and enhancing the supply of electron donors and other cofactors [18,19,20], are also widely used in the construction of microbial cell factories.

Considering the long pathway for the synthesis of ERD, the synthetic pathway from glucose to ERD was organized into the following three modules (Figure 1). Module 1: the synthesis of tyrosine and phenylalanine from glucose. Then, tyrosine ammonia lyase (FjTAL), phenylalanine ammonia lyase (AtPAL), cinnamate-4-hydroxylase (AtC4H), cytochrome P450 reductase 2 (AtATR2), and cytochrome b5 (Cyb5) catalyze the synthesis of *p*-coumaric acid (*p*-CA) from tyrosine and phenylalanine [21]. Module 2: (2*S*)-naringenin (NAG) was synthesized by *p*-CA under the combined action of 4-coumarate CoA ligase (Pc4CL), chalcone synthase (PhCHS), and chalcone isomerase (MsCHI) [22]. Module 3: ERD is characterized by an additional hydroxylation modification at the 3′ position of the B ring compared with NAG, and the flavonoid 3′-hydroxylase (F3′H) and AtATR2 catalyze this reaction [23].

As early as 2000, Schoenbohm et al. identified a flavonoid 3′-hydroxylase (F3′H) from *Arabidopsis thaliana* and synthesized ERD through the addition of NAG [24]. To achieve a higher conversion of NAG, Gao et al. identified and optimized the promoters of *SmF3*′*H* in combination with *SmCPR* from *Silybum marianum* in *S. cerevisiae*. A directed evolution approach was employed to further improve the conversion of ERD by *SmF3*′*H*/*SmCPR*, resulting in 3.3 g L^−1^ ERD in a 5 L fermenter, and the conversion rate was 62.0% [25]. Besides *S. cerevisiae*, some researchers have also explored the possibility of using other microorganisms to produce ERD. In *Escherichia coli*, Zhu et al. addressed the low cytochrome P450 expression by expressing a truncated flavonoid 3′-hydroxylase (t*F3*′*H*) and a truncated P450 reductase (t*CPR*), and the production of ERD from _L_-tyrosine reached 107 mg L^−1^ [26]. In *Yarrowia lipolytica*, Lv et al. determined the optimal gene copy numbers for *CHS* and *CPR*, and further enhanced the precursor supply by expressing genes associated with chorismate and malonyl-CoA; the engineered strain was able to produce 134.2 mg L^−1^ of ERD [19]. There are still many limiting factors for the efficient synthesis of ERD. Overall, more efficient methods for ERD production should be developed.

In this study, we report establishing a de novo ERD-producing *S. cerevisiae* platform based on regulating the balance of metabolic flux. First, we evaluated different promoters for the expression of pathway genes in the *TAL* branch and *PAL* branch. The best strain could produce 523.86 mg L^−1^ of *p*-CA. Subsequently, the efficient synthesis of NAG was achieved using promoter engineering and terminator engineering, further confirming the important effect of appropriate promoters on NAG synthesis. Finally, by increasing the copy number of *ThF3*′*H* in the engineered strain, an ERD production level of 132.08 mg L^−1^ was achieved. This work lays the foundation for the cost-effective production of hydroxyflavonoids using *S. cerevisiae* as an engineered strain, which expands our ability to synthesize drugs and natural products using natural organisms.

## 2. Materials and Methods

Chemicals and Reagents: Ampicillin was obtained from Solarbio Technology (Beijing, China). The plasmid mini extraction kit, gel extraction kit, and one-step cloning kit were purchased from Vazyme Bio (Nanjing, China). The primer synthesis and DNA sequencing were performed by Sango Bio (Shanghai, China). All the flavonoid standards used in the assay were purchased from Yuanye Bio (Shanghai, China).

Plasmid and Strain Construction: *E. coli* JM109 was used for the construction of all plasmids. The *S. cerevisiae CEN.PK2-1D* derivative (C800-Δubi4, *MATα, ura3-52*, *trp1-289*, *leu2-3*,*112*, *his3Δ1*, *MAL2-8^c^*, *SUC2*, *gal80::G418*) was used for the expression of heterologous pathways. The plasmids and strains used in this study are listed in Appendix A. C800 genomic DNA was used as a template for the amplification of the yeast endogenous promoters, genes, and terminators. Plasmids or synthetic fragments were used as DNA templates for the amplification of optimized heterologous genes. All fragments were ligated into vectors using the one-step cloning kit. The CRISPR/Cas9 system tool was used for gene integration. CRISPR/Cas9 specific guide RNAs were set up using the website http://www.rgenome.net/cas-designer/ (accessed on 1 June 2023) and constructed using the Golden Gate Assembly Kit. The integration sites were derived from the C800 genomic DNA and primers synthesized by Sangon Biotech, and the integration fragments were constructed using fusion PCR. All the plasmids and integration sites were confirmed using Sanger sequencing (Sangon Biotech).

All the primers used for all constructs are listed in Appendix A. Detailed information about the heterologous genes used in this study is listed in Appendix A. The sequences of the natural strong terminators and artificial strong terminators used in this paper are listed in Appendix A.

Culture Media and Conditions: *E. coli* JM109 and the *E. coli* seed cultures were cultivated in Luria–Bertani (LB) medium (10 g of L^−1^ tryptone, 10 g L^−1^ NaCl, and 5 g L^−1^ yeast extract). Terrific Broth (TB; consisting of 24 g L^−1^ yeast extract, 12 g L^−1^ tryptone, 4 mL L^−1^ glycerol, 9.4 g L^−1^ K_2_HPO_4_, 2.2 g L^−1^ KH_2_PO_4_) was used for the *E. coli* fermentation. To select or maintain the plasmids, the *E. coli* was supplemented with 100 mg of L^−1^ ampicillin. The *S. cerevisiae* cultivation was performed using the YNB medium (1.74 g L^−1^ yeast nitrogen base without amino acids, 5 g L^−1^ ammonium sulfate, and 20 g L^−1^ glucose. When necessary, 50 mg L^−1^ leucine, 50 mg L^−1^ histidine, 50 mg L^−1^ tryptophan, or 50 mg L^−1^ uracil was added to the medium). The *S. cerevisiae* fermentation was performed using the YPD medium (20 g L^−1^ peptone, 10 g L^−1^ yeast extract, and 20 g L^−1^ glucose).

*E. coli* fermentation: Single colonies of *E. coli* were picked and inoculated into 2 mL of LB medium and incubated overnight at 37 °C and 220 rpm, inoculated into 250 mL shake flasks supplemented with 30 mL of TB at an inoculum of 1% (*v*/*v*), and incubated at 37 °C and 220 rpm. When the OD_600_ reached 0.6–0.8 after 3–4 h of incubation, IPTG at a final concentration of 0.2 mM was added, and this was incubated for 24 h at 23 °C and 220 rpm.

*S. cerevisiae* fermentation: A single colony of *S. cerevisiae* was picked and inoculated into 3 mL of the YNB medium and incubated at 220 rpm and 30 °C for 24 h. Then, the yeast OD_600_ was detected, and the yeast was inoculated at a final concentration of 0.05 into 250 mL shake flasks spiked with 30 mL of YPD and incubated at 30 °C for 96 h and 220 rpm.

Co-culture system analysis: For the co-culture of *S. cerevisiae* and *E. coli*, the process of *S. cerevisiae* seed culture and fermentation was the same as that for *S. cerevisiae* alone. At 36 h of *S. cerevisiae* fermentation, single colonies of *E. coli* were inoculated into 2 mL of LB medium overnight at 37 °C. The single colonies were transferred into 30 mL of TB medium at an inoculation rate of 1% (*v*/*v*) and cultured for 3–4 h at 37 °C until the OD_600_ reached 0.6–0.8. Strategy 1 is that the *E. coli* fermentation broth was centrifuged at 8000 rpm for 2 min, and the collected *E. coli* cells were resuspended with 1 mL of YPD. Then, we added the cells to the yeast’s fermentation broth, and IPTG was also added at a final concentration of 0.2 mM. The *E. coli* and yeast were co-cultured for 96 h at 23 °C and 30 °C, respectively. Strategy 2 is that the *E. coli* was induced first by adding IPTG at a final concentration of 0.2 mM for 24 h at 23 °C. Then, the cells were collected and suspended in 1 mL of fresh YPD medium, and the resuspended *E. coli* was added to the yeast broth and co-cultivated at 30 °C for 24 h.

Metabolite extraction and quantification: Briefly, 0.5 mL of cell culture was mixed with an equal volume of absolute ethanol (100% *v*/*v*), fully rotated, and then centrifuged at 13,000× *g* for 5 min. The supernatant was stored at −20 °C for the HPLC analysis. The supernatant was quantified using an LC-20AD prominence instrument (Shimadzu, Kyoto, Japan) equipped with a diode array detector and an InertSustain C18 250 mm × 4.6 mm column (particle size 5 μm; Shimadzu, Kyoto, Japan). The flavonoids were detected at 284 nm. At a column temperature of 30 °C, the metabolites of 10 µL of supernatant were separated. The samples were analyzed using two solvent gradients: water with 0.1% formic acid (A) and acetonitrile (B). The flow rate was 1 mL/min. The procedure was started with 10% solvent A, after which part of it was linearly increased (0–10 min) from 10 to 40%, 40 to 50% (10–20 min), decreased to 10% for 5 min (20–25 min), and finally maintained at 10% for 5 min (25–30 min).

## 3. Results

### 3.1. Engineering of the Upstream Pathway Enables the Production of p-Coumaric Acid

As shown in Figure 1, a single heterologous enzymatic step of the _L_-tyrosine pathway (the *TAL* branch) or two enzymatic steps of the _L_-phenylalanine pathway (the *PAL* branch) was used for the de novo synthesis of *p*-CA. For the *TAL* branch, a highly specific tyrosine ammonia lyase from *Flavobacterium johnsoniae* (FjTAL) was integrated into chromosomal loci XI-2 (Figure 2a) [27] using promoters *GAL7*p or *INO1*p (Figure 2a), respectively, resulting in strains SQ01 and SQ02 with titers of 81.56 mg L^−1^ and 38.46 mg L^−1^ *p*-CA, respectively (Figure 2b,c). For the *PAL* branch, phenylalanine ammonia lyase (AtPAL2), cinnamate-4-hydroxylase (AtC4H), and cytochrome P450 reductase (AtATR2) from *A. thaliana* and cytochrome B5 from *S. cerevisiae* (Cyb*5*) were integrated into chromosomal loci XI-5 (Figure 2a), resulting in strains SQ03 and SQ04. As shown in Figure 2b,c, strains SQ03 and SQ04 produce 200.82 mg L^−1^ and 511.96 mg L^−1^ of *p*-CA, respectively. It was clear that the *PAL* branch was much more efficient for the production of *p*-CA compared to the *TAL* branch, even though the intracellular _L_-tyrosine concentration is usually slightly higher than the _L_-phenylalanine concentration under glucose-limited conditions [21]. To enhance the *p*-CA production, we further introduced *FjTAL* with the promoter *GAL7*p into strain SQ04. This resulted in increased *p*-CA production, up to 523.86 mg L^−1^ in strain SQ05 (Figure 2b,c), so it was chosen for the de novo synthesis of NAG.

### 3.2. Promoter-Library-Based (2S)-Naringenin Pathway Optimization

Promoters are fundamental transcriptional regulatory elements that control the quantitative and temporal regulation of protein expression and have been widely used to fine-tune the gene expression in pathway engineering in *S. cerevisiae* [28]. Promoter strength refers to the efficiency and effectiveness of a promoter in regulating the transcriptional activity of a gene at a particular time. Since the expression of genes by promoters in *S. cerevisiae* is subject to stringent metabolic regulation, promoters with the same strength at a particular time will also show different strengths at other times due to differences in metabolic regulation, which will lead to differences in the production of the final products [28]. Therefore, for the synthesis of NAG, a suitable promoter is essential.

To further construct the NAG biosynthetic pathway, we first screened *Pc4CL*, *PhCHS*, and *MsCHI* from different combinations of promoters and tested them in the wild-type strain C800 (*CEN.PK2-1D, gal80::kanMX*) (Figure 3a,b). A total of 22 plasmids were constructed and then introduced into strain C800, resulting in strains N01–N22. Among the above 22 strains, the NAG production varied from 0 to 176.84 mg L^−1^, and the one containing pY26-*ADH1*t-*PhCHS*-*GAL10*p-*ALD5*p-*MsCHI*-*ADH2*t-*ARO7*p-*Pc4CL*-*CYC1*t was found to be best. Strain N08 showed the highest NAG production when *p*-CA was added (Figure 3d). This finding confirmed the importance of the gene expression balance in pathway engineering.

Next, we introduced the plasmid pY26-*ADH1*t-*PhCHS*-*GAL10*p-*ALD5*p-*MsCHI*-*ADH2*t-*ARO7*p-*Pc4CL*-*CYC1*t into strain SQ05, resulting in strain N23. As shown in Figure 3c,e, strain N23 produced 133.66 mg L^−1^ of NAG with 247.60 mg L^−1^ of *p*-CA as an intermediate. Compared with the exogenous addition of *p*-CA to strain N08, the NAG synthesized from glucose had a lower conversion rate from *p*-CA.

### 3.3. The Half-Life of mRNA Is Regulated Based on Terminator Optimization

Terminators are genetic elements that terminate transcription independently of the encoding gene and are important control elements in synthetic biology [29]. Studies have shown that terminators vary from strong to weak and can directly affect the amount of mRNA, the stability of the mRNA, the cleavage efficiency, and the polyadenylation reaction, as well as the length and stability of the 3′-UTR region, which ultimately affect the level of gene expression [30].

In our study, artificial terminators (Tsynth27 [29], SynTer8 [31], T-1300 stem 6 [30]), natural strong terminators (*HOG1*t [30], *CPS1*t [32], *DIT1*t [33]), and commonly used terminators (*CYC1*t, *ADH1*t, *ADH2*t) were selected as the expression elements for *Pc4CL*, *PhCHS*, and *MsCHI*, resulting in strains N24-N29 (Figure 4a,b). When the terminator of *Pc4CL* was replaced with Tsynth27 or *HOG1*t, the final production of NAG was increased by about 10% (N24, N25). When the terminator of *PhCHS* was replaced with *CPS1*t, the NAG production was further increased from 133.66 mg L^−1^ to 149.59 mg L^−1^ in strain N27 (Figure 4b). Based on strains N25 and N27, we replaced the original *Pc4CL* terminator with *HOG1*t and the original *PhCHS* terminator with *CPS1*t, resulting in strain N30. As shown in Figure 4c, the production of *p*-CA was further decreased, but the NAG production decreased slightly. The original terminator is more suitable for NAG production, suggesting that simply increasing the strength of the terminator sometimes has a negative effect, and a more fine-grained regulation of the metabolic process is needed.

### 3.4. Copy Number Regulation Improves (2S)-Eriodictyol Production

NAG and ERD are structurally similar, differing only by one hydroxyl group at the 3′ position of the B ring, and NAG is able to synthesize ERD when catalyzed by the flavonoid 3′-hydroxylase from *Tricyrtis hirta* (*ThF3*′*H*) [34]. As shown in Figure 5a, a new peak appeared in the ERD standards and was identified as ERD. However, we constructed strain N31 to harbor the pRS424 plasmid expressing *ThF3*′*H*, and 175.39 mg L^−1^ of *p*-CA and 73.19 mg L^−1^ of NAG remained in the fermentation broth (Figure 5b). To further promote the conversion of the precursors into ERD, we increased the expression of *ThF3*′*H* from one to two copies on the plasmid pRS425 and generated strain N32 (Figure 5b). Surprisingly, although the NAG level remained at 72.09 mg L^−1^, the production of ERD was significantly increased from 98.64 mg L^−1^ to 132.08 mg L^−1^, and the production of *p*-CA was significantly decreased to 109.42 mg L^−1^ (Figure 5b). These results suggest that the synthesis of ERD should be facilitated by strategies such as screening for a more active F3′H, screening for a more suitable promoter, or constructing an electron channel for efficient electron transfer between the P450 and reductase.

### 3.5. Biosynthesis of (2S)-Hesperetin from Glucose by Co-Culturing the Engineered S. cerevisiae and E. coli Strains

(2*S*)-hesperetin has a potential therapeutic role in the prevention and mitigation of diabetes and its complications. ERD is converted into (2*S*)-hesperetin by the flavonoid 4′-*O*-methyltransferase (Figure 6a). In this study, we introduced the flavonoid 4′-*O*-methyltransferase (*MpF4*′*OMT*) from *Mentha × piperita.* We first evaluated the activity of *MpF4*′*OMT* on different substrates, NAG and ERD, in *S. cerevisiae* and *E. coli*. As shown in Figure 6c,d, in *S. cerevisiae*, *MpF4*′*OMT* converted 32.68% NAG and 6.06% ERD, respectively, whereas in *E. coli*, *MpF4*′*OMT* converted more than 80% NAG and 80% ERD, suggesting that *E. coli* is a more suitable host for expressing *MpF4*′*OMT*. So, we tested the de novo synthesis of (2*S*)-hesperetin by co-culturing *S. cerevisiae* and *E. coli* (Figure 6b). As shown in Figure 6e, only a small amount of (2*S*)-hesperetin was produced in the fermentation broth, reaching 14.50 mg L^−1^. The experiments showed that we obtained a low content of (2*S*)-hesperetin, but a more effective co-culture method may be found through further investigation.

## 4. Discussion

Our study has shown that one of the major bottlenecks in the ERD production pathway is the conversion of NAG into ERD. Although the strain could convert 1.50 g L^−1^ of NAG to obtain ERD on the gram scale at the shake flask level in Gao’s study [25], there is only one step to convert NAG into ERD. In our fermentation process, even large amounts of the intermediates, NAG and *p*-CA, could not effectively be converted into ERD, maybe due to the intermediates being secreted out of the cells before being converted into the downstream product [35]. Therefore, in addition to focusing on the rapid conversion of intermediates, rapid synthesis of the downstream products also reduces the accumulation of intermediates [36].

Previously, Liu used minimal medium to ferment strains QL01 and QL13, resulting in 337.6 mg L^−1^ and 12.9 mg L^−1^ of *p*-CA, respectively [21]. However, when we used the same genes and constructed strains SQ01 and SQ04 for *p*-CA production, the titers we achieved were 81.56 mg L^−1^ and 511.96 mg L^−1^, respectively, and this difference may be due to the different medium, promoters, terminators, and integration sites. Our results also confirmed the importance of using suitable promoters for *p*-CA synthesis, and the use of *GAL* series promoters increased the *p*-CA titer 2.12-fold and 2.55-fold, respectively, compared with the other promoters. In addition, as shown in Figure 2b, a small amount of cinnamic acid was detected in the fermentation broth of the SQ05 strain, but pinocembrin was not detected in Figure 3c, which may be due to the fact that the cinnamic acid content was so low that it was consumed during the transformation process.

The NAG synthesis pathway is a rate-limiting step compared to the synthesis of *p*-CA. As shown in Figure 3e, still 247.60 mg L^−1^ of *p*-CA could not be converted into NAG even after the promoter was optimized. According to the literature, although CHS is the rate-limiting step in NAG synthesis [37,38], there is a need to find a higher catalytic activity of CHS or balance the metabolic flux by screening more suitable promoter pairings to increase the NAG production. In addition, some unwanted byproducts were produced, as shown in Appendix A, and this disruption of CHS catalysis adversely affects the efficiency of flavonoid biosynthesis due to the derailment of the chalcone production pathway [39]. Meanwhile, the endogenous double-bond reductase *TSC13* in the yeast also led to the accumulation of phloretic acid through the reduction of *p*-coumaroyl-CoA [40]. *PAD1* and *FDC1* from *S. cerevisiae* are thought to be cinnamic acid decarboxylases that convert trans-cinnamic acid into styrene [41]. Thus, they also catalyze the decarboxylation of *p*-CA [42].

ERD is a potent source of natural remedies for maintaining a high standard of health. In recent years, many strategies have been employed to improve ERD production, but the conversion has not been very high. In our study, we increased the ERD production by increasing the copy number of *ThF3*′*H* (Figure 5b). Li et al. improved the protein solubility in *E. coli* by removing the hydrophobic N-terminal residues that serve as a membrane anchor in the endoplasmic reticulum, obtaining a higher conversion rate [16]. Liu et al. screened a more suitable cytochrome P450 reductase (CPR) to transfer electrons. And strategies such as the deletion of *ROX1* and *HXM1* to increase the level of the cofactor heme, the deletion of *OPI1* and overexpression of *INO2* to amplify the endoplasmic reticulum, and enabling a higher protein-folding capacity were used to improve the catalytic efficiency of P450 [12]. Although there are still many strategies to further improve the final production of ERD, our study demonstrated that a higher expression of *F3*′*H* was an important factor to promote the metabolic flux toward the downstream pathway.

At the same time, we evaluated the catalytic activity of *MpF4*′*OMT* in *S. cerevisiae* and *E. coli* and found that it was better for (2*S*)-hesperetin production in *E. coli*. Therefore, a co-culture method was employed. The final titer of (2*S*)-hesperetin was quite low, but more efforts are needed to further increase the (2*S*)-hesperetin level produced by optimizing the ratios of *S. cerevisiae* and *E. coli* in subsequent studies.

In conclusion, this study reveals the importance of regulating the heterologous gene expression for the efficient production of ERD. We optimized the synthetic pathway from glucose to NAG based on the properties of different promoters and terminators and achieved a titer of 132.08 mg L^−1^ of ERD by increasing the copy number of *ThF3*′*H*. Although the catalytic efficiency of ThF3′H needs to be further improved for more efficient synthesis of ERD, this study emphasizes the importance of the balanced utilization of metabolic intermediates and the controlled expression of genes in the biosynthetic pathway.

## Figures and Tables

**Figure 1 jof-10-00119-f001:**
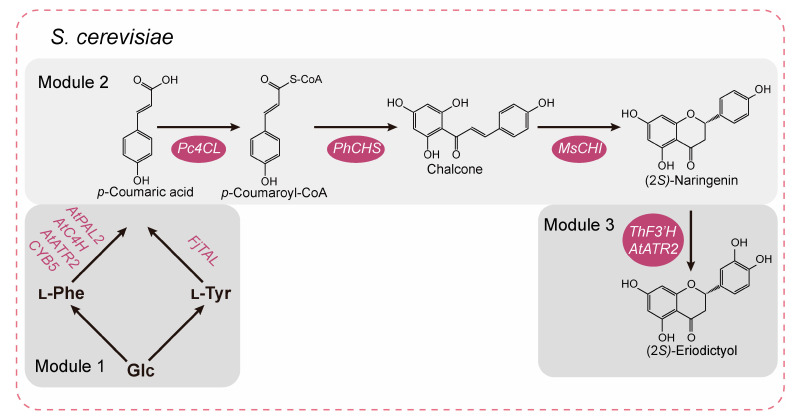
Pathway for the de novo synthesis of (2*S*)-eriodictyol in *S. cerevisiae*. FjTAL, tyrosine ammonia lyase from *Flavobacterium johnsoniae*; AtPAL2, phenylalanine ammonia lyase 2 from *A. thaliana*; AtC4H, cinnamate-4-hydroxylase from *A. thaliana*; AtATR2, cytochrome P450 reductase 2 from *A. thaliana*; Cyb5, cytochrome b5 from *S. cerevisiae*; Pc4CL, 4-coumarate CoA ligase from *Petroselinum crispum*; PhCHS, chalcone synthase from *Petunia × hybrida*; MsCHI, chalcone isomerase 1 from *Medicago sativa*; ThF3′H, flavonoid 3′-hydroxylase from *Tricyrtis hirta*.

**Figure 2 jof-10-00119-f002:**
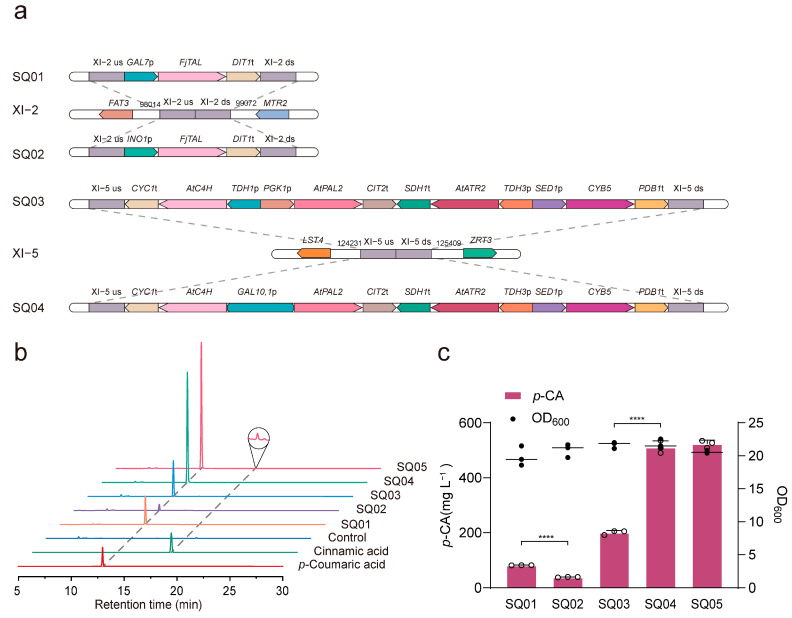
Reconstruction of the *p*-CA synthesis pathway in *S. cerevisiae*. (**a**) Integration mapping of *FjTAL*, *AtPAL2*, *AtC4H*, *AtATR2*, and *CYB5* to the *S. cerevisiae* genome. (**b**) HPLC profiles of *p*-CA-producing strains, *p*-CA standard, and cinnamic acid standard. (**c**) *p*-CA production, as well as OD_600_ readings, of strains SQ01-SQ05. Statistically significant differences between two data groups were determined using a two-tailed Student’s *t*-test (one-tailed; two-sample unequal variance; **** *p* < 0.0001). All data represent the mean of *n* = three biologically independent samples, and error bars show standard deviation.

**Figure 3 jof-10-00119-f003:**
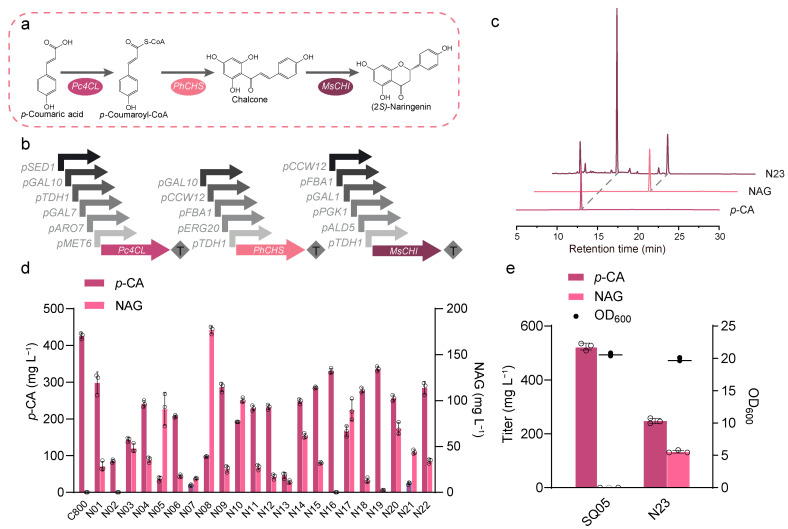
Reconstruction of the NAG biosynthetic pathway in *S. cerevisiae*. (**a**) The NAG synthesis pathway from *p*-CA in *S. cerevisiae*. This is followed by the conversion of *p*-CA into NAG via three enzymatic steps: 4-coumaroyl-CoA ligase (Pc4CL), chalcone synthase (PhCHS), and chalcone isomerase 1 (MsCHI). (**b**) Promoter–ORF combinations used in the combinatorial promoter library for fine-tuning the expression of enzymes in the NAG biosynthetic pathway. (**c**) HPLC profile of strain N23. (**d**) The effect of different combinations of promoters in the NAG biosynthetic pathway on NAG production with the supplementation of *p*-CA as a precursor. (**e**) Final results of construction of the de novo biosynthetic pathway for NAG in *S. cerevisiae*. *p*-CA and NAG production as well as OD_600_ readings for strain N23. All data represent the mean of *n* = three biologically independent samples, and error bars show standard deviation.

**Figure 4 jof-10-00119-f004:**
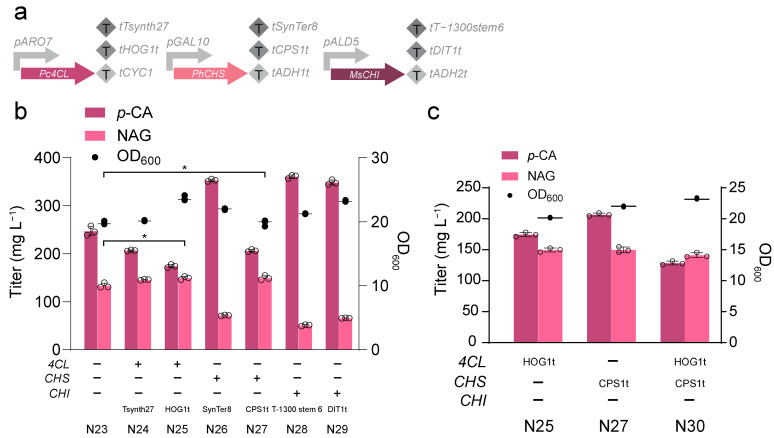
Optimization of the terminators in NAG biosynthetic pathway. (**a**) Terminator combinations used in this study for tuning half-life of NAG biosynthetic pathway gene mRNA. (**b**) NAG production when terminators are replaced separately in NAG biosynthetic pathway. (**c**) Optimization of different combinations of terminators in NAG biosynthetic pathway. “−” indicates that the terminators used by the genes corresponding to the left side of “−” have not been replaced, and “+” indicates that the terminators used by the genes corresponding to the left side of “+” have been replaced with the terminators below “+”. Statistically significant differences between two data groups were determined using a two-tailed Student’s *t*-test (one-tailed; two-sample unequal variance; * *p* < 0.05). All data represent the mean of *n* = three biologically independent samples, and error bars show standard deviation.

**Figure 5 jof-10-00119-f005:**
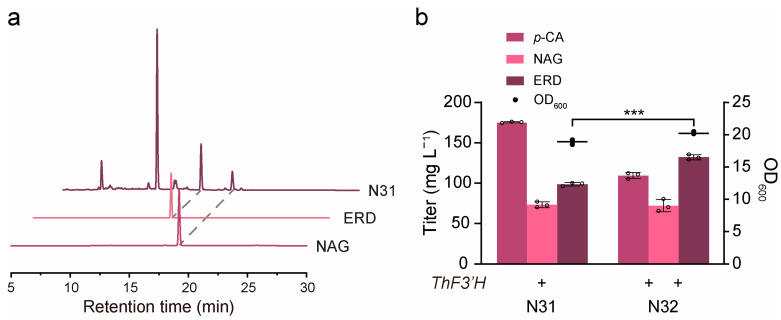
Construction of the de novo biosynthetic pathway for ERD. (**a**) HPLC profile of N31. (**b**) Results of flask cultures of the N31 and N32 strains are shown. “+” means *ThF3*′*H* expressing one copy number and “++” means *ThF3*′*H* expressing two copies. Statistically significant differences between two data groups were determined using a two-tailed Student’s *t*-test (one-tailed; two-sample unequal variance; *** *p* < 0.001). All data represent the mean of *n* = three biologically independent samples, and error bars show standard deviation.

**Figure 6 jof-10-00119-f006:**
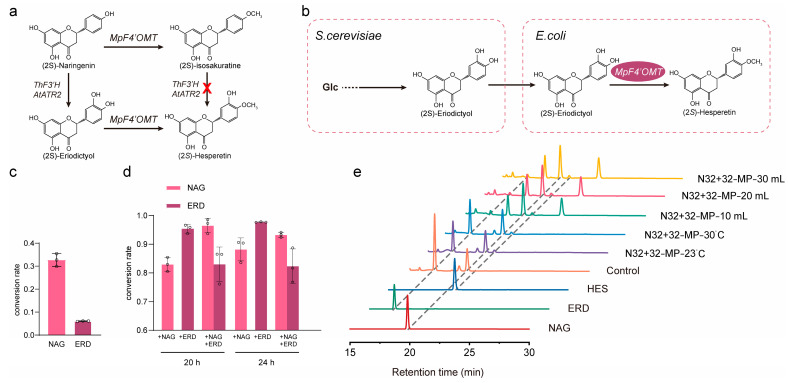
Biosynthesis of (2*S*)-hesperetin by co-culturing the engineered *S. cerevisiae* and *E. coli* strains. (**a**) (2*S*)-hesperetin synthesis pathway. Red crosses indicate blocked metabolic pathways. (**b**) (2*S*)-hesperetin synthesis pathway in co-culture of *S. cerevisiae* and *E. coli*. (**c**) Efficiency of conversion of NAG and ERD according to expression of *MpF4*′OMT in *S. cerevisiae*. (**d**) Expression of *MpF4*′*OMT* in *E. coli* and conversion with the addition of NAG or/and ERD. (**e**) HPLC profiles of *S. cerevisiae* and *E. coli* co-cultures. All data represent the mean of *n* = three biologically independent samples, and error bars show standard deviation.

## Data Availability

The data supporting the reported results can be found in the Appendix A.

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
