# Peer review of "Systems Metabolic Engineering of Saccharomyces cerevisiae for the High-Level Production of (2S)-Eriodictyol"

_jof, 2024, doi:10.3390/jof10020119_

Round 1

Reviewer 1 Report

Comments and Suggestions for Authors

The manuscript "Systems Metabolic Engineering of Saccharomyces cerevisiae for High-level Production of (2S)-Eriodictyol" describes the construction and optimization of yeast S.cerevisiae strain for the production of valuable 2D-eriodictyol by fermentation. The manuscript is clearly wrtten, and the results, matherials and methods are adequately presented. In their work the authors combined genes the wanted to express with various promoter and terminator sequences to obtain the maximum yield of the desired compound. Their work reviels the need for experimental testing of various combinations of regulatory sequences for each synthetic metabolic  pathway one would want to express in vivo, since the level of desired product depends on the balance of expression levels of multiple genes. This is a valuable insight for synthetic biology in general. There aro no major issues, and I can recommend this article for publication. 

Comments on the Quality of English Language

Of minor issues, I recommend rechecking the English, as minor mistakes and typos in langauge are present.

Reviewer 2 Report

Comments and Suggestions for Authors

The manuscript by Zhang et al. describes the use of S. cerevisiae to produce the medically relevant flavonoid (2S)-eriodictyol. The authors engineer pathways for de novo synthesis of ERD, and demonstrate optimization of strain construction for particular modules of the overall pathway. The final engineered yeast strain yielded very high levels of ERD. The authors also make an initial attempt to co-culture S. cerevisiae and E. coli to convert ERD produced by yeast into (2S)-hesperetin using enzymes expressed in E. coli. Overall, this work is of good quality and suitable for publication in Journal of Fungi. I have some minor comments that the authors should address before the manuscript is accepted, though.

Minor comments

1. Section 3.4: The authors conclude this section with the statement that there should be optimization of the promoter for F3’H, but it is surprising that they did not screen for more suitable promoters. The authors screened for promoters for expressing other enzymes in the pathway, so it is unclear if there is any reason why they did not do so for F3’H. Finding a more suitable promoter that further increases yield of ERD would improve the impact of the study. If the authors do not carry out such optimization, it would be helpful to address if there is any particular reason that they did not try to optimize that final enzymatic step.

2. Section 3.5: Similar to the comment above, just one quick attempt is made to see if co-culturing of yeast and E. coli to convert ERD to (2S)-hespertin is made. Even an initial attempt to optimize the co-culturing that yields moderately improved results could improve the paper. If the authors do not wish to do that, it would be important to at least explain the value of optimizing a co-culture, as opposed to obtaining ERD from the yeast fermentation and separately fermenting E. coli to convert that ERD to (2S)-hespertin.

3. In Figure 2a, it would be helpful to show a nearby genomic feature on either side of each of the chromosome XI integration sites so that readers familiar with the yeast genome will know where integration is occurring.

4. It would help to clarify what the authors state in lines 191-193. For instance, are the authors trying to state that two equally strong promoters could result in different product yields because those different promoters will respond differently as the cultures grow? Or do they mean that it is hard to predict the expression of an enzyme from a given promoter because expression will change as the culture grows?

5. For Figure 5B, I don’t think it will be clear to all readers that the plus symbol on the x-axis is indicating copy number of F3’H, especially because no reference is made to the symbol in the legend. It would help to clearly show that there are different copy numbers visually.

6. In the discussion, it would be helpful to suggest possible changes to the yeast genome that might help reduce problems with intermediates being diverted into other molecules and lowering final yield of ERD.

7. The numbers in the references section are given on the left margin but then also at the start of each reference. It would be better to just have the numbers on the margin.

Comments on the Quality of English Language

The writing overall is good, there are just minor typos or confusing phrases that the authors can find using spellcheck and a grammar check. Below are some specific examples:

Line 44: “serious accumulation” would be better as toxic accumulation or substantial accumulation (or another similar word)

Line 44: replace “finial” with “final”

Line 46: Do the authors mean in “different ways” rather that “different directions”?

Lines 58 and 62: Don’t capitalize synthesis in the middle of these sentences

Line 73: S. cerevisiae is a microorganism, so the authors should refer to the possibility of “other” microorganisms or restate the second half of the sentence in another way that makes it clear that S. cerevisiae is also a microorganism.

Line 84: Replace “stain” with “strain”

Line 100: Replace “were” with “was”

Line 263: I am not sure why there is an X between the genus and species names. Is that supposed to be there?

Line 272: The phrase “needed to be found in the following study” would read better as “may be found through further investigation” or a similar phrase

Reviewer 3 Report

Comments and Suggestions for Authors

The reviewed paper represents results of an interesting, comprehensive study at a very high level. The amount of work is striking and results are of obvious practical importance. The main drawback of the paper is a large amount of mistakes. I tried to point most of them further:

14 - “obesity effects” - maybe “antiobesity effects”

18 - “and the intermediate metabolites were reduced.” - maybe “the intermediate metabolites levels were reduced.”

21 - “ThF3′H from Tricyrtis hirta“ - maybe “ThF3′H gene from Tricyrtis hirta“.

21 – “Finally, the production of ERD was further increased, reaching 132.08 mg L-1 by increasing the copy  number of ThF3′H” – better rephrase, e.g. “Finally, by multiplying the copy number of ThF3′H gene the production of ERD was further increased, reaching 132.08 mg L-1”.

34 – “… making it become …” – rephrase.

41 and further – “Expression of heterologous proteins …” – although I now see it in many papers, I was always taught that the term “expression” should be used for the genes. For proteins it is better to say production, or synthesis.

58 “… from glucose to p-coumaric acid (p-CA), Synthesis of tyrosine and phenylalanine from glucose.”

59 “Then, tyrosine and phenylalanine are synthesized to p-CAcatalyzed by tyrosine…”

61-62 “Module 2, Synthesis of (2S)-naringenin (NAG) from p-CA was under the combined action of 4- coumarate CoA ligase…”

72 “on a 5L fermenter” maybe “in a 5L fermenter”

72-73 “Besides in S. cerevisiae, some researchers have also explored the possibility of microorganisms to produce ERD.” – maybe “Besides S. cerevisiae, some researchers have also explored the possibility of other microorganisms to produce ERD.”

77 “determined that the optimal gene copy numbers” – maybe “determined the optimal gene copy numbers”

80-81 “Overall, more efficient methods for ERD production should be attempted.” – maybe “Overall, more efficient methods for ERD production should be developed.”

87-88 “Finally, the engineered strain was able to produce 132.08 mg L-1 ERD by increasing the copy number of ThF3'H.” – rephrase, maybe “Finally, the by increasing the copy number of ThF3'H in engineered strain ERD production level of 132.08 mg L-1 was achieved.”

93 “FjTAL” and other names – usually for S. cerevisiae genes are marked with italics and enzyme names are written in regular font starting from capital letter (e.g. Pho5).

100 Ampicillin were…”

111 “…heterologous genes).

129 “…to the medium.).

146-148 “Option 1 was to collect E. coli at 8000 rpm and added E. coli to the fermentation 146 broth of S. cerevisiae. Then we added IPTG at a final concentration of 0.2 mM, and fer- 147 mented for 96 h at 23°C and 30°C, respectively. Option 2 was added 0.2 mM IPTG, and E. 148 coli was incubated at 23°C for 24 h.”

166-167 “…a single heterologous enzymatic step of L-tyrosine (the TAL branch) or two enzymatic steps of L-phenylalanine (the PAL branch)…” – maybe “…a single heterologous enzymatic step of L-tyrosine pathway (the TAL branch) or two enzymatic steps of L-phenylalanine pathway (the PAL branch)…”

191 “Since the promoter strength of yeast varies with the growth and development of yeast…” – maybe “Since the strength of yeast promoters varies with the growth and development of yeast culture…”

200 “…resulting in production of NAG in the presence of 426.54 mg L-1 of p-CA in strain N08 (Figure 3d).” – please rephrase.

213 – “(d) HPLC profile of strain…”

215 – “(e) Construction of the de novo biosynthetic 215 pathway for NAG in S. cerevisiae.” - This phrase does not describe the (e) graph. Maybe “Final results of construction…”

218 – “Terminators function as gene elements independent of coding genes to terminate transcription are an important control element in synthetic biology [29].” – rephrase.

222 – “…affect the degree of gene expression [30].” – better “…affect the level of gene expression [30].”

224 and further – “(Tsynth27 [29]SynTer8 [31]T-1300 stem 6 [30])” – strange commas

229 – “When the terminator of PhCHS was replaced to CPS1t, NAG production was further increased from 133.66 mg L-1 to 149.59 mg L-1 in strain N27 (Figure 230 4b).” – was such difference statistically significant? Which statistical criteria were used to analyze this?

243 – “Compared with the structures of ERD and NAG, there was only one more hydroxyl  group at the 3' position of the B ring, and NAG could be able to be converted to ERD by flavonoid 3′-hydroxylase from Tricyrtis hirta (ThF3’H) [36].” – please, rephrase.

247 – “However, expressing ThF3′H on plasmid pRS424 in strain N24 still had 175.39 mg L-1 p-CA and 73.19 mg L-1 NAG left in strain N31 (Figure 5b).” It is not clear, which strain was used. Please rewrite.  

250-252 “Surprisingly, although NAG remained at 72.09 mg L-1 , the production of ERD was significantly increased from 98.64 mg L-1 to 132.08  mg L-1 and the production of p-CA was significantly decreased to 109.42 mg L-1 (Figure 5b).” These results should be supported with proper statistical analysis.

257 “Increasing the copy numbers of ThF3′H to improve ERD production.” – this is not an informative description of the graph 5b.

263 “Mentha × piperita” – why x?

270 “The  experiments showed that although (2S)-hesperetin we obtained was quite low, a more effective co-culture method needed to be found in the following study.” – please, rephrase.

276 “(c) Expression of MpF4′OMT in S. cerevisiae, conversion with the addition of NAG and ERD.”- not an informative description for the graph.

282 “…in Gao’s study [25],there is only…”

283 “In our fermentation process, still amounts of intermediates…” – “maybe still large amounts”

285 “Therefore, we need to focus on the conversion of intermediates and also the rapid synthesis of the  downstream products is conducive to the reduction of intermediate product accumulation [35, 36].” – please, rephrase.

I do not feel competent enough to correct English in the paper, but it seems to me that careful grammar checking is needed.

Round 2

Reviewer 1 Report

Comments and Suggestions for Authors

On my opinion, the authors took in to consideration all of the issues in the manuscript and the manuscript can be accepted in the present form.

Reviewer 2 Report

Comments and Suggestions for Authors

The revised manuscript by Zhang et al. addresses all the minor concerns that I raised. I think that the manuscript is now suitable for publication in JoF.

Reviewer 3 Report

Comments and Suggestions for Authors

I accept the corrections and answers to my questions.